# Profiles of Internet Use and Health in Adolescence: A Person-Oriented Approach

**DOI:** 10.3390/ijerph18136972

**Published:** 2021-06-29

**Authors:** Henri Lahti, Nelli Lyyra, Lauri Hietajärvi, Jari Villberg, Leena Paakkari

**Affiliations:** Faculty of Sport and Health Sciences, University of Jyväskylä, P.O. Box 35, FI-40014 Jyväskylä, Finland; nelli.lyyra@jyu.fi (N.L.); lauri.hietajarvi@helsinki.fi (L.H.); jari.j.villberg@jyu.fi (J.V.); leena.paakkari@jyu.fi (L.P.)

**Keywords:** internet, internet use, adolescent, health

## Abstract

(1) Background: Internet use has become an integral part of adolescents’ daily lives. It is important to understand how adolescents use the internet, and how this use is associated with demographic factors and health from a person-oriented perspective. (2) Methods: The study applied the Finnish nationally representative HBSC data (persons aged 11, 13, and 15, n = 3408), descriptive observation, latent class analysis, and multinomial logistic regression analysis. (3) Results: Entertainment activities (listening to music) and socially oriented activities (liking posts, talking online) were the most prevalent among adolescents, but gender differences emerged. Five different internet user profiles were identified (encompassing interest-driven, friendship-driven, abstinent, irregular, and excessive users). Interest-driven users participated in interest- and media-oriented activities. Adolescents in the interest-driven user group were more likely to be boys and participants with low academic achievement, high parental monitoring, and high problematic social media use. Friendship-driven users participated in socially oriented activities. Adolescents in the friendship-driven user group were more likely to be girls and participants aged 13 or 15, with high peer and family support. Abstinent users participated only in entertainment, while irregular users showed no particularly high involvement in any internet activity. Adolescents in the abstinent and irregular user groups were likely to be boys and participants aged 11 with high family support. Excessive users had high involvement in internet activities overall. Adolescents in the excessive user group were more likely to be participants with high problematic social media use and were most likely to feel low and tired on school mornings. (4) Conclusion: The study confirmed the prevalence of internet use. It identified five internet user profiles and differences between user profiles regarding individual and social factors and health outcomes.

## 1. Introduction

Internet use has increased enormously in recent decades, leading society into the digital era. Today’s adolescents, Generation Z, are the first generation with widespread access to the internet at an early age, and they have an unprecedented amount of technology in their upbringing [1]. Hence, Generation Z has been described as the “net generation” [2]. Since internet use has become a significant part of adolescent everyday life, there are significant questions concerning how adolescents’ internet use is associated with individual and social factors, and with possible health implications [3]. So far, most studies have approached adolescents’ internet use by seeking to establish general relationships. In contrast, this study adopted a person-oriented approach. In so doing, it aimed to go beyond mere consideration of the average experiences of adolescents and to explore “the interindividual variability and complexity that is a hallmark of human growth” [4].

There is a growing consensus that internet use is a complex and multidimensional phenomenon. Recent review studies have concluded that the effects of internet use on adolescent health depend on various factors, notably including the type of use. In previous research, different types of internet users have been identified through patterns of participation in different kinds of internet activities, with the patterns being labeled as “genres” [5], “typologies” [6], and “profiles” [7]. In a year-long ethnographic investigation performed on adolescents aged 12–19, Ito et al. [5] identified three “genres” of internet use, namely (i) friendship-driven “hanging out” (motivated by the desire to maintain connections with friends), (ii) interest-driven “messing around” (motivated by fortuitous searching and experimental play), and (iii) creatively oriented “geeking out” (intensive commitment to and engagement with technology, often involving one particular medium, genre, or type of technology *or* creative production *or* gaming). In a meta-analysis, Brandtzæg [6] identified eight “typologies” of internet use. The typologies differed according to frequency of use, variety of use, activities participated in, and platforms used; the user types were divided into non-users (no internet use), sporadics (low internet use, oriented towards no particular activity), debaters (medium internet use, oriented towards blogs and social networks), entertainment users (medium internet use, oriented towards new media and video games), socializers (medium internet use, oriented towards social networks), lurkers (medium internet use, oriented towards social networks and new media), instrumental users (medium internet use, oriented towards shopping online), and advanced users (intensive internet use, oriented towards all activities). In further research, conducted on Finnish adolescents from elementary, secondary, and high school, Hietajärvi et al. [7] identified six “participation profiles”, consisting of social-networking-oriented participation (oriented towards communicating with friends), knowledge-oriented participation (oriented towards sharing and gaining information related to one’s interests), media-oriented participation (oriented towards to the long term and to complex activities such as creating and sharing videos, pictures, and music), action gamers (oriented towards first-person shooter games, role-playing games, and adventure games), social gamers (oriented towards playing games with social motives such as fun and exercise) and (among high school students), separate blogging-oriented participators (oriented towards activities relating explicitly to blogging).

In addition to the type of use, studies suggest that internet use and its effect on adolescents’ health is driven by individual-level factors such as gender, age [8], family affluence [9], and fear of missing out [10]. Also important are friend-level factors (peer support) [11] and family-level factors (parental surveillance) [9,12], in addition to contextual factors (notably a culture of surveillance and comparison) [13]. Overall, longitudinal research has suggested that the effects of the internet differ from adolescent to adolescent [3].

Previous studies have identified both benefits and drawbacks regarding adolescents’ internet use, and there has been no clear consensus. On one hand, internet use has been associated with benefits such as new and profound means of self-exploration, self-reflection [14], increased social capital [15], social support and opportunities for finding friends [16], learning and creativity [17], access to information [18], and promotion of self-esteem, social competence, and empathy [19]. On the other hand, meta-analytic studies have highlighted associations between internet use (especially excessive and problematic use) and negative health outcomes in adolescence, including psychosomatic complaints (such as depressive symptoms and anxiety) [20,21,22] and lower sleep quality [23,24]. 

Despite the recent increase in research on adolescent internet use, gaps remain, which could be filled by approaches addressing the multidimensional, interindividual complexity of adolescents’ internet use. In addition, it is important to study internet use and its relation to individual factors, social factors, and health outcomes via a person-oriented approach, applied to a nationally representative sample. In employing such a person-oriented approach, this study is one of the few to tap into the subject from a multidimensional, interindividual standpoint, going beyond purely aggregate experiences [4]. The research questions for the study were:

What is the prevalence of different internet activities among adolescents, and are there differences in terms of gender? (RQ1)What kind of internet user profiles can be identified, and how are they different in terms of participation in internet activities? (RQ2)How are various individual factors (gender, age, family affluence, health literacy, academic achievement) and social factors (friend support, family support, parental monitoring) associated with internet user profiles? (RQ3)How are health outcomes (self-rated health, feeling low, morning tiredness) and problematic social media use associated with internet user profiles? (RQ4)

## 2. Materials and Methods

### 2.1. Design and Participants

The data used in this study were collected as part of a cross-national collaborative study called Health Behavior in School-Aged Children (HBSC). The present study involved 3408 Finnish adolescents aged 11 years (*n* = 993), 13 years (*n* = 1246), and 15 years, *n* = 1169). The sample included boys (*n* = 1706) and girls (*n* = 1702). The schools were chosen using a cluster sampling method aimed at overall reliability, bearing in mind that the schools should be nationally representative in terms of size and the municipalities in which they were located. The participants were asked to fill in a self-completed questionnaire. Administration took place within the classroom. The data collection followed guidelines on the responsible conduct of research according to the protocol of the international HBSC study [25].

### 2.2. Measures

Self-reported gender and age were measured by asking adolescents to select the correct alternative.

*Internet activity* was measured via 16 items on how often adolescents participated in the following internet activities [26]: read or look at content (browse), “dig” or “give thumbs up” to other people’s postings (like), listen to music (listen), read or look at what acquaintances are doing (follow), write a blog or other text (blog), look for information (info), comment on interesting things (comment), share different content (share), tell acquaintances what I am doing (post), take or edit pictures (picture), play games (game), get to know new people (know people), look for like-minded company (company), take or edit videos (video), make or edit music (music), and talk on the internet (e.g., via WhatsApp or Skype) (talk). The questionnaire employed a Likert-type scale ranging from 1 to 6 (1 = never, 2 = less than once a week, 3 = once a week, 4 = several days a week, 5 = every day once a day, and 6 = several times every day).

*The Family Affluence Scale III* [27] was used to measure self-reported socioeconomic position. FAS III includes six items: number of family computers, number of family bathrooms, ownership of a car, ownership of a dishwasher, having one’s own bedroom, and number of family vacations during the past 12 months. The computed scores were recoded into three categories to indicate relative family affluence: low family affluence (lowest 20%), medium family affluence (middle 60%), and high family affluence (highest 20%), according to the HBSC protocol [25]. *Parental monitoring* was measured via a six-item four-point scale covering adolescents’ perceptions of parental monitoring and awareness [28] regarding where they go after school, free-time activity, going out at night, internet activity, spending money, and friends. Scores covering monitoring by both mother and father were computed to form a sum score that was then recoded into three categories: low parental monitoring (lowest 33.3%), medium parental monitoring (middle 33.3%), and high parental monitoring (highest 33.3%).

*Health literacy* was measured using the Health Literacy for School-Aged Children (HLSAC) instrument [29,30]. The scale consists of ten items that assess the knowledge and competencies that promote health among adolescents. The responses were totaled to produce a sum score, which was then categorized into one of three groups: low health literacy (values 10–25), medium health literacy (values 26–35), and high health literacy (values 36–40). [31]. 

*Academic achievement* was measured by asking students to indicate their most recent marks on first language and mathematics. The responses ranged from 4 (fail) to 10 (excellent). The mean value for both marks was calculated and recoded into one of three categories: low academic achievement (4–7), medium academic achievement (7.5–8.5), and high academic achievement (9–10) [32].

*Peer support* [33] was measured via a multidimensional scale consisting of four items covering friends’ help, being able to count on friends, emotional support, and talking about problems with friends. The scale ranged from 1 = very strongly disagree to 7 = very strongly agree. The score was calculated by adding the items together and dividing them by four. Computed scores were recoded to low peer support (1–2.9), medium peer support (3–5), and high peer support (5.1–7). *Family support* [33] was measured via a multidimensional scale consisting of four items: family help, emotional support, talking about problems with family, and family’s willingness to help in making decisions. The scale ranged from 1 = very strongly disagree to 7 = very strongly agree. The score was calculated by adding the items together and dividing them by four. Computed scores were recoded to low family support (1–2.9), medium family support (3–5), and high family support (5.1–7). 

*Self-rated health* (SRH) was measured by a single question on the individual’s perception and evaluation of his or her health [34], and the response options were poor, fair, good, and excellent. Response options fair and poor were combined to indicate low SRH, with good and excellent indicating high SRH. *Feeling low* was measured via a HBSC symptom checklist (HBSC-SCL) [35]. Respondents evaluated the frequency of their feeling low over the last six months. Feeling low weekly or more often was categorized as feeling low frequently.

*Morning tiredness* was measured with a single item: “How often do you feel tired when you get up on school mornings?” [36]. The response categories were rarely or never, sometimes, 1–3 times a week, and 4 or more times a week. Being tired four or more times a week was categorized as a risk for adolescent health.

*Problematic social media use* (PSMU) was measured via the nine-item Social Media Disorder Scale (SMD-scale) using a dichotomous (No/Yes) answer scale [37]. Based on the values obtained, the respondents were categorized into three groups: a no-risk group, a moderate risk group (at heightened risk of developing problematic use), and a problematic use group. The cut-off value for the problematic use group was 6 or more “yes” answers, for the moderate risk group it was 2–5 “yes” answers, and for the no-risk group it was 0–1 “yes“ answers [38].

### 2.3. Analyses

Descriptive analyses were used to explore the prevalence of internet activities among adolescents. Cross-tabulation, chi-square χ²-test, and confidence intervals (95% CI) were used to explore the differences in internet activities between boys and girls. 

### 2.4. Mixture Model Selection and Multinomial Logistic Regression

In general, a benefit of mixture models is the variety of fit indices available to examine the best fitting profile solution. However, simulation studies have shown that none of the indices alone can provide a reliable way to detect the proper solution across all combinations of, for instance, model specification, sample size, or possible indicators [39,40,41]. The model considered here was a latent class analysis (LCA) with categorical indicators, regarding which Nylund, Asparouhov, and Muthén [42] suggest the Bayesian information criterion (BIC) and the bootstrapped likelihood ratio test (BLRT) as the best indicators overall. For models similar to those potentially applicable in the present study, the simulations also indicated that the CAIC and Vuong–Lo–Mendell–Rubin (VLMR) likelihood ratio test would perform well. Yang [43] suggests sample-size-adjusted (N* = (N + 2)/24) BIC (aBIC) to be the overall best-performing indicator, with good performance also noted for sample-size-adjusted consistent Akaike’s information criterion (aCAIC) when the number of participants per class was lower (but at least *n* ≥ 50 for aBIC and *n* ≥ 84 for aCAIC). In the Yang [43] simulations, BIC and CAIC were also shown to have satisfactory accuracy when the sample sizes were higher. Morovati [44] suggests use of BIC, aBIC, and CAIC when the sample size is >1000, and both aBIC and BLRT otherwise. On the basis of the simulations mentioned above, we utilized both BIC and CAIC in evaluating the number of classes, and VLMR and BLRT in comparing neighboring models. Lower values of BIC and CAIC pointed towards a better fit to the data. 

Furthermore, to limit computational time, and to avoid capitalizing on chance over too many statistical tests, we decided on class enumeration via a two-fold process [42]. First of all, we examined the range of plausible solutions with BIC and CAIC by increasing the number of classes until the lowest value (or an elbow point [40,43]) was identified. Secondly, we tested between competing neighboring models using VLMR and BLRT. In addition, we relied on entropy value as an indicator of the classification quality, with entropy > 0.8 indicative of a clear classification of participants into their most likely classes. Importantly, given the discrepancies between statistical information criteria across situations, we relied heavily on the interpretability of the additional classes in terms of revealing qualitative differences in the shape of the profiles, rather than mere level differences [39,44].

Multinomial logistic regression was used to examine the associations between internet user profiles, individual and social factors, health outcomes, and problematic social media use. The strength of the association was indicated by odds ratio (OR) values. A listwise deletion procedure was used to handle missing data. The significance level was set at *p* < 0.05. Descriptive statistics and multinomial logistic regression analyses were conducted with IBM SPSS Statistics version 26 (IBM, Armonk, NY, USA), and latent class analysis with Mplus version 8.5 (Muthén & Muthén, Los Angeles, CA, USA).

## 3. Results

### 3.1. The Prevalence of Internet Activities and Association with Gender (RQ1)

The most prevalent internet activities were listening to music (43.0%), liking posts (40.4%), and talking online (40.2%) (Table 1). Among boys, the most common activities were listening to music (36.3%), talking online (35.9%), and playing games (35.5%), and among girls, liking posts (49.9%), listening to music (49.4%), and talking online (44.3%). Only two of the internet acti vities (browsing and blogging) were not significantly associated with gender.

### 3.2. Identification of Internet User Profiles and Differences between Internet User Profiles Regarding Internet Activities (RQ2)

In identifying internet user profiles via LCA, the information criterion BIC suggested up to nine and CAIC up to eight classes. By contrast, VLMR did not show support for increasing the number of classes above five (Table 2). BLRT showed nonconvergence and was not considered. The entropy value was high (>0.85) for all solutions considered. Based on the substantive information provided by the five-class solution, we ended up with five internet user profiles (interest-driven users (*n* = 302), friendship-driven users (*n* = 1163), abstinent users (*n* = 574), irregular users (*n* = 799), and excessive users (*n* = 354). 

#### 3.2.1. Interest-Driven Users

Interest-driven users were reflected through having at least regular but moderate engagement (at least weekly to multiple times a week) in all internet activities (Figure 1, Figure A1). Thus, they were overall among the most digitally active groups. What distinguished this class was that they reported the highest probability of engaging regularly (even several times a week) in creative and media-oriented activities, such as editing videos, and making and editing music.

#### 3.2.2. Friendship-Driven Users

Friendship-driven users demonstrated moderate to high engagement (from at least several times a week up to several times a day) in socially oriented activities such as liking, talking online, following, commenting, and posting (Figure 1, Figure A1). As regards other activities, friendship-driven users did not engage in creative and media-oriented activities such as blogging, taking and editing videos, and making and editing music. At the same time, they exhibited low engagement regarding in search of like-minded company and getting to know new people. What distinguished this class from others was their high involvement in socially oriented activities, in contrast to low engagement in creative activities.

#### 3.2.3. Abstinent Users

Abstinent users were reflected through their generally low engagement (from never to once a week) in internet activities. Abstinent users were, in general, the least active group regarding internet use, except for their involvement in listening to music, playing video games, and talking online (Figure 1, Figure A1).

#### 3.2.4. Irregular Users

Irregular users reported no particularly high engagement in any internet activity; however, they showed more variation in their activity than abstinent users. The irregular users reported low to moderate engagement (less than once a week to several times a week) in socially oriented activities such as liking, talking online, following, commenting, sharing, and posting, and also in interest-driven activities such as browsing and searching for info (Figure 1, Figure A1). Irregular users also reported moderate engagement (at least several times a week) in playing video games. What distinguished irregular users from other internet user profiles was their erratic participation in most internet activities.

#### 3.2.5. Excessive Users

Excessive users formed the most active internet user profile with at least moderate, often excessive involvement (at least daily to several times a day) in many internet activities, including liking, following, commenting, sharing, posting, talking online, searching for information, and playing video games (Figure 1, Figure A1). Excessive users regularly got to know people and looked for people with similar interests. They were the most active group in taking and editing pictures.

### 3.3. Internet User Profiles Associated with Individual and Social Factors (RQ3), Health Outcomes, and Problematic Social Media Use (RQ4)

In the sample, the most normative internet user profile was that of friendship-driven users (36.4%) (Table 3). All the variables, except self-rated health, were associated with internet user profiles. Regarding individual factors, gender differences were found. Girls were more likely to be friendship-driven users, whereas boys were more likely to be interest-driven, abstinent, and irregular users. Adolescents aged 13 and 15 years old were more likely to be friendship- and interest-driven users, whereas 11 year olds were more likely to be abstinent and irregular users. Participants with high health literacy were most likely to be excessive users and adolescents with high academic achievement were most likely to be friendship-driven users. Adolescents with low academic achievement were most likely to be interest-driven users.

As regards social factors, participants with high peer support were more likely to be friendship-driven and excessive users, whereas adolescents with high family support were more likely to be irregular, friendship-driven, and abstinent users. Adolescents with high parental monitoring were most likely to be interest-driven users.

In terms of health outcomes, self-rated health was not associated with the internet user profiles. However, adolescents feeling low and tired on school mornings were most likely to be excessive users. Adolescents feeling low less than weekly were most likely to be abstinent and irregular users.

Participants belonging to the problematic social media user group were most likely to be interest-driven and excessive users, whereas participants belonging to the moderate risk group were most likely to be excessive and friendship-driven users.

Table 4 presents the results of the multinomial logistic regression. The friendship-driven user group was used as the reference group, as it was the most normative. As regards individual factors, boys were four times more likely to be interest-driven users and almost three times more likely to be abstinent or irregular users than to be friendship-driven users (Table 4). Adolescents aged 11 years old were three times more likely to be abstinent users and over three times more likely to be irregular users than friendship-driven users. Participants with low health literacy were almost three times more likely to be abstinent users compared to the reference group. Moreover, adolescents with low academic achievement were over four times more likely to be interest-driven users than friendship-driven users. As regards social factors, adolescents with low peer support were almost three times more likely to be abstinent users compared to the reference group. Participants belonging to the problematic social media user group were over three times more likely to be interest-driven users and almost three times more likely to be excessive users compared to friendship-driven users.

## 4. Discussion

Using a nationally representative sample from Finland, the study elucidated the prevalence of adolescents’ internet use. It identified five internet user profiles and analyzed how these were related to individual and social factors, health outcomes, and problematic social media use. The study represents one of the few to employ a person-oriented approach, approaching the matter from an interindividual standpoint.

Internet use is common among adolescents, with 45% of teens being almost constantly online [14]. In general, adolescents spent most time engaging in entertainment (listening to music) and in socially oriented activities (talking online, liking) and less time on complex and technically demanding activities (taking and editing videos, making and editing music). These findings are supported also by the findings of the Pew Research Center [14]. Due to the accessibility of smartphones and the development of the information society, it has become possible for adolescents to stay constantly connected [45] and to carry their entire entertainment libraries in their pockets; thus, they can engage in social and entertainment-oriented activities more frequently. As regards gender, our study suggests that socially oriented activities (liking, talking online, following, commenting, posting) are more common among girls, whereas video gaming and media-oriented activities (taking and editing videos, making and editing music) are more common among boys.

Five internet user profiles were identified: interest-driven users, friendship-driven users, abstinent users, irregular users, and excessive users. The profile structures, which reflected the genres of participation identified by Ito et al. [5] and the typologies studied by Brandtzæg [6], were somewhat similar to the profiles of Hietajärvi et al. [7]. Interest-driven use resembled the media-oriented participation identified by Hietajärvi et al. [7], and was reflected through more complex activities related to creating and sharing media (pictures, videos, music). In contrast, friendship-driven use in this study resembled the social networking-oriented participation identified by Hietajärvi et al. [7], or the friendship-driven “hanging out” described by Ito et al. [5], motivated by the desire to communicate with friends through social media. Abstinent users and irregular users were contrasted with the category of “sporadics” identified by Brandtzæg [6]. Note, however, that within our study, irregular users were more versatile in their use than abstinent users, who leaned more towards “non-users” except in terms of some forms of entertainment (listening to music and playing games). Excessive users participated intensively in internet activities, apart from complex, media-oriented activities and blogging. In previous studies, the most excessive user group often overlapped with the most advanced user group in terms of the complex nature of the preferred activities [6], with activities also linked to other activities such as high engagement in social media and in interest-related searching. This makes the excessive user group identified in the present study somewhat different, insofar as they were less engaged in technically demanding activities. Note also that in our study, video gaming was measured by only one item. This may have had an effect on the profile structures, bearing in mind that in the study by Hietajärvi et al. [7] (for example), two video gaming-related user profiles were identified, namely “action gaming” and “social gaming”.

The internet user profiles differed in terms of individual and social factors, health outcomes, and the prevalence of problematic social media use. Beyens [3] found that the effects of internet use differed between individual adolescents, and this appeared to also be the case in our study. Moreover, it has been suggested that schools may tend to alienate digitally engaged students [46], and this was supported by our finding that adolescents in the interest-driven user group were most likely to be students with low academic achievement. It should be noted that low academic achievement might also be explained through an energy-depletion process related to an imbalance between adolescents’ resources and the demands of schoolwork [47]. In contrast, adolescents in the friendship-driven user group were likely participants with high peer support and were more likely to be girls, a finding in line with the study conducted by Inchley [9] wherein girls were more likely than boys to communicate with friends online. Given that the benefits of internet use include increased social capital [15] and social support [16], friendship-driven use may overall be beneficial for adolescent health.

The profiles encompassing lesser participation in internet activities (including abstinent and irregular users) were more likely to be found among 11 year olds. The developmental level might be an explanatory factor in the age distribution of the profiles, insofar as younger adolescents have had less time to experience the different forms of internet use.

The model of compensatory internet use theorizes that the negative outcomes related to internet use may be due to attempts to escape real life [48]. In contrast, Valkenburg and Peter [49] argue that the effects of the internet are based on individual susceptibilities. In our study, the negative health outcomes (“feeling low” and “being tired on school mornings”) were more common among active participants in internet activities (the excessive users and the interest- and friendship-driven users) than among abstinent and irregular users. Negative health outcomes were most common among excessive users; nevertheless, self-rated health was not associated with the profiles.

The adolescents with problematic social media use were three times more likely to be interest-driven users and almost three times more likely to be excessive users, in comparison with the friendship-driven users (Table 4). Overall, the evidence indicates that moderate to high socially oriented internet use does not intrinsically predict problematic social media use. However, compared to more passive user profiles (the abstinent and irregular users), the actively participating profiles were more likely to belong to the at-risk and problematic social media user groups. Persons working with young people should be adept at identifying content and qualitative differences in internet use and the various contexts of use, since the intensity of use is not the only predictor of health outcomes or of problematic social media use.

The present study has several strengths. For instance, we employed a large, nationally representative sample in conjunction with a person-oriented approach. This offered multiple benefits, bearing in mind that the majority of studies on adolescent internet use have so far been variable-oriented. Our approach thus sheds new light on the phenomenon, with the individual taken as the unit of the analysis. Another strength of the study is the use of internationally validated variables. However, the study had certain limitations. First, it can be argued that self-report instruments may not give a sufficiently objective view of adolescents’ internet activity, due to the risk of their overly emphasizing the amount of activity [50]. Second, the intensity scale—ranging from never participating to participating several times a day—may not have given sufficient information on the intensity of internet use. Use of the internet several times a day has in fact become the status quo; in this sense, using this criterion as a measurement of intensive use creates the risk of falsifying the results. In future studies, one could seek to use objective measurements of the time and frequency of internet use in addition to the content and quality of screen time. The tools would include objective measurement of time spent online via smartphone application tracking apps, detailed time-diary methods, or repeated-experience sampling methods. In addition, longitudinal research on the direction of the association between internet user profiles and health outcomes should be studied.

## 5. Conclusions

The study accomplished its objectives in terms of using a person-oriented approach to study the prevalence of adolescents’ internet use. Entertainment activities and socially oriented activities were the most prevalent among adolescents, but gender differences emerged. Additionally, the study successfully identified five different internet user profiles (encompassing interest-driven, friendship-driven, abstinent, irregular, and excessive users). The study also confirmed differences between the internet user profiles in terms of individual and social factors, health outcomes, and problematic social media use. In the future, we suggest that objective measurement tools such as smartphone application tracking apps could be used to gain more detailed insights into the qualitative and quantitative aspects of adolescents’ internet use. Furthermore, longitudinal research on the direction of the association between internet user profiles and health outcomes should be conducted.

## Figures and Tables

**Figure 1 ijerph-18-06972-f001:**
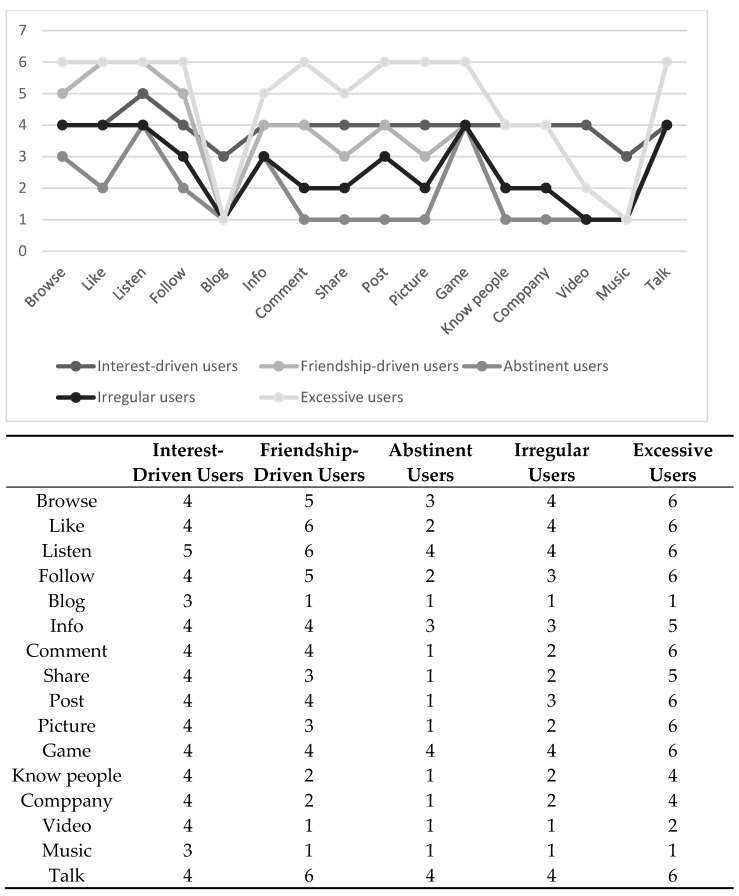
Medians for internet activities within the internet user profiles (1 = never, 2 = less than once a week, 3 = once a week, 4 = several days a week, 5 = every day once a day, and 6 = several times every day).

**Table 1 ijerph-18-06972-t001:** Prevalence of internet activities in total and by gender.

	All	Boys	Girls	
	Several Times a Day % [95 CI]	Several Times a Day % [95 CI]	Several Times a Day % [95 CI]	χ^2^(df); *p*-Value
Browse	29.4 [28.0–31.2]	28.0 [25.8–30.3]	30.7 [28.3–33.0]	χ^2^(6) = 6.3; 0.281
Like	40.4 [38.7–41.9]	30.3 [28.0–32.5]	49.9 [47.5–52.5]	χ^2^(6) = 147.0; <0.001
Listen	43.0 [41.3–44.7]	36.3 [33.8–38.7]	49.4 [47.0–51.8]	χ^2^(6) = 68.5; <0.001
Follow	23.2 [21.7–24.6]	16.4 [14.4–18.3]	29.6 [27.3–32.0]	χ^2^(6) = 135.6; <0.001
Blog	1.4 [1.0–1.9]	1.5 [0.9–2.2]	1.4 [0.9–1.9]	χ^2^(6) = 1.7; 0.891
Info	9.6 [8.7–10.8]	10.8 [9.3–12.3]	8.5 [7.1–9.8]	χ^2^(6) = 16.2; 0.006
Comment	9.7 [8.6–10.6]	8.2 [6.9–9.6]	11.0 [9.4–12.6]	χ^2^(6) = 22.9; <0.001
Share	7.6 [6.7–8.5]	6.4 [5.1–7.6]	8.7 [7.3–10.0]	χ^2^(6) = 17.9; 0.003
Post	12.1 [10.8–13.1]	8.3 [6.9–9.6]	15.6 [13.8–17.4]	χ^2^(6) = 89.1; <0.001
Picture	9.8 [8.7–10.9]	6.8 [5.5–8.1]	12.6 [10.9–14.3]	χ^2^(6) = 172.4; <0.001
Game	22.4 [21.1–23.9]	35.5 [33.0–38.0]	10.1 [8.7–11.7]	χ^2^(6) = 630.7; <0.001
Know people	4.1 [3.4–4.8]	5.5 [4.4–6.6]	2.8 [2.0–3.6]	χ^2^(6) = 86.7; <0.001
Company	3.7 [3.4–4.8]	4.2 [3.1–5.2]	3.2 [2.3–4.0]	χ^2^(6) = 39.4; <0.001
Video	3.2 [2.6–3.8]	3.8 [2.9–4.8]	2.6 [1.9–3.3]	χ^2^(6) = 60.3; <0.001
Music	2.1 [1.6–2.6]	2.6 [1.8–3.4]	1.6 [1.1–2.2]	χ^2^(6) = 77.2; <0.001
Talk	40.2 [38.4–41.8]	35.9 [33.4–38.3]	44.3 [41.9–46.7]	χ^2^(6) = 32.4; <0.001

**Table 2 ijerph-18-06972-t002:** Information criterion values of latent class analysis for different internet profile solutions.

	Parameters	LL	BIC	CAIC	Entropy	VLMR
1 class	80	−77,577.33	155,800.12	155,880.74		
2 classes	161	−73,627.99	148,554.94	148,717.19	0.84	
3 classes	242	−71,704.57	145,361.62	145,605.51	0.83	
4 classes	323	−70598.05	143,802.09	144,127.61	0.87	0.00
5 classes	404	−69,711.77	142,683.05	143,090.21	0.86	0.00
6 classes	485	−68,914.44	141,741.91	142,230.69	0.88	0.82
7 classes	566	−68,317.94	141,202.41	141,772.83	0.89	
8 classes	647	−67,802.68	140,825.41	141,477.46	0.88	
9 classes	728	−67,438.32	140,750.22	141,483.90	0.87	
10 classes	809	−67,131.18	140,789.44	141,604.76	0.87	

**Table 3 ijerph-18-06972-t003:** Internet user profiles association with individual factors, social factors, health outcomes, and problematic social media use.

		Interest-Driven Users (*n =* 302)	Friendship-Driven Users (*n =* 1163)	Abstinent Users (*n =* 574)	Irregular Users (*n =* 799)	Excessive Users (*n =* 354)	
		% [95% CI]	% [95% CI]	% [95% CI]	% [95% CI]	% (95% CI)	χ^2^(df); *p*-Value
All		9.5 [8.4–10.5]	36.4 [34.8–38.2]	18.0 [16.7–19.3]	25.0 [23.7–26.5]	11.1 [10.0–12.2]	
Gender	Girl	32.1 [26.8–37.5]	66.5 [63.9–69.2]	41.8 [37.6–45.9]	43.6 [40.2–47.1]	52.3 [47.2–57.7]	χ^2^ (4) = 190.3; <0.001
	Boy	67.9 [62.5–73.2]	33.5 [30.8–36.1]	58.2 [54.1–62.4]	56.4 [52.9–59.8]	47.7 [42.3–52.8]	
Age	15	40.1 [34.8–45.4]	39.0 [36.1–41.8]	27.5 [24.0–31.2]	23.5 [20.7–26.4]	39.1 [34.0–44.5]	χ^2^ (8) = 143.5; <0.001
	13	35.4 [30.1–40.7]	40.0 [37.1–43.0]	32.6 [28.7–36.2]	35.7 [32.3–38.9]	37.7 [32.6–42.8]	
	11	24.5 [19.9–29.5]	21.1 [18.7–23.4]	39.9 [36.1–43.9]	40.8 [37.3–44.4]	23.2 [18.7–27.5]	
Family affluence	High	18.6 [14.5–23.1]	19.2 [16.9–21.6]	16.2 [13.1–19.5]	16.0 [13.4–18.7]	24.7 [20.6–29.7]	χ^2^ (8) = 35.2; <0.001
	Medium	57.9 [52.1–63.4]	63.1 [60.1–66.2]	56.8 [52.6–61.0]	59.2 [55.4–62.5]	54.7 [49.1–59.6]	
	Low	23.4 [18.6–28.6]	17.8 [15.5–20.0]	27.0 [23.1–30.6]	24.8 [22.2–28.0}	20.6 [16.3–25.0]	
Health literacy	High	28.1 [21.4–35.2]	39.2 [36.1–42.6]	29.0 [23.9–34.0]	26.0 [21.5–30.4]	49.4 [42.6–55.7]	χ^2^ (8) = 77.17; <0.001
	Medium	57.1 [49.5–63.8]	55.1 [51.7–58.5]	56.6 [50.8–62.0]	66.3 [61.6–71.3]	43.4 [37.0–49.4]	
	Low	14.8 [10.2–19.9]	5.7 [4.1–7.4]	14.5 [10.8–18.9]	7.7 [5.9–10.4]	7.2 [3.8–10.6]	
Academic achievement	High	13.9 [9.6–18.8]	33.1 [30.3–36.2]	22.9 [18.7–27.4]	27.9 [23.7–31.8]	26.7 [22.1–31.8]	χ^2^ (8) = 67.6; <0.001
	Medium	44.2 [37.5–51.0]	47.7 [44.4–51.1]	46.4 [40.7–51.5]	47.3 [42.7–51.9]	43.8 [38.0–49.6]	
	Low	41.8 [35.1–48.6]	19.2 [16.4–21.7]	30.7 [25.6–36.1]	24.8 [20.7–28.8]	29.5 [24.0–35.3]	
Peer support	High	56.0 [49.4–61.8]	74.9 [72.3–77.5]	58.9 [54.5–63.2]	65.3 [61.8–68.6]	73.1 [67.6–77.9]	χ^2^ (8) = 69.5; <0.001
	Medium	31.7 [26.3–37.1]	18.8 [16.5–21.1]	27.5 [23.8–31.4]	24.8 [21.9–27.9]	18.3 [14.1–22.8]	
	Low	12.4 [8.1–17.0]	6.3 [4.9–7.7]	13.6 [10.7–16.7]	9.9 [7.7–12.1]	8.7 [5.8–12.2]	
Family support	High	58.9 [53.1–64.9]	74.9 [72.1–77.5]	73.0 [68.5–76.8]	76.9 [73.7–80.0]	68.4 [62.9–73.5]	χ^2^ (8) = 40.5; <0.001
	Medium	28.7 [23.0–34.0]	17.9 [15.7–20.2]	18.1 [14.9–21.6]	15.8 [13.2–18.6]	24.2 [19.4–29.4]	
	Low	12.5 [8.7–16.6]	7.2 [5.8–8.9]	8.9 [6.6–11.4]	7.3 [5.3–9.2]	7.4 [4.5–10.6]	
Parental monitoring	High	44.5 [37.4–52.2]	28.5 [25.2–32.2]	30.6 [25.0–35.9]	33.5 [28.9–38.1]	34.1 [27.7–40.5]	χ^2^ (8) = 25.9; <0.001
	Medium	30.2 [23.6–36.8]	34.6 [31.2–38.1]	31.3 [25.7–36.6]	36.6 [31.7–41.5]	29.1 [23.2–35.0]	
	Low	25.3 [19.2–31.9]	36.9 [33.8–40.5]	38.0 [32.4–43.7]	29.9 [25.3–34.5]	36.8 [30.0–43.2]	
Self-rated health	Good	84.4 [80.1–88.4]	86.2 [83.9–88.0]	86.4 [83.6–89.2]	86.4 [84.0–88.6]	81.0 [76.8–85.3]	χ^2^ (4) = 7.3; 0.123
	Poor	15.6 [11.6–19.9]	13.8 [12.0–16.1]	13.6 [10.8–16.4]	13.6 [11.4–16.0]	19.0 [14.7–23.2]	
Feeling low	Less than	64.2 [58.9–69.9]	61.2 [58.6–63.9]	75.6 [72.1–78.7]	72.3 [69.2–75.7]	53.8 [48.4–59.2]	χ^2^ (4) = 73.6; <0.001
	More than	35.8 [30.1–41.1]	38.8 [36.1–41.4]	24.4 [21.3–27.9]	27.7 [24.3–30.8]	46.2 [40.8–51.6]	
Tired on school mornings	Less than	66.6 [61.3–71.9]	66.3 [63.6–69.2]	74.0 [70.2–77.7]	75.5 [72.2–78.6]	63.2 [58.4–68.0]	χ^2^ (4) = 32.6; <0.001
	More than	33.4 [28.1–38.7]	33.7 [30.8–36.4]	26.0 [22.3–29.8]	24.5 [21.4–27.8]	36.8 [32.0–41.6]	
Social media use	No risk	44.3 [38.1–50.2]	51.6 [48.7–54.5]	73.1 [69.1–76.8]	63.8 [60.3–67.3]	38.4 [32.8–44.0]	χ^2^ (8) = 231.2; <0.001
	moderate risk	33.2 [27.7–38.4]	39.9 [37.1–42.7]	22.1 [18.7–25.8]	32.1 [28.7–35.5]	43.4 [38.1–48.7]	
	Problematic	22.5 [17.6–27.7]	8.5 [7.0–10.2]	4.7 [3.1–6.5]	4.1 [2.8–5.5]	18.2 [14.1–22.0]	

**Table 4 ijerph-18-06972-t004:** Multinomial logistic regression on the associations between internet user profiles, individual and social factors, health outcomes, and problematic social media use using friendship-driven users as the reference group.

	Interest-Driven Users	Abstinent Users	Irregular Users	Excessive Users
	OR [95% CI]	OR [95% CI]	OR [95% CI]	OR [95% CI]
Sex				
Girls	1	1	1	1
Boys	4.06 [2.99–5.50]	2.73 [2.17–3.45]	2.58 [2.09–3.18]	1.93 [1.48–2.53]
Age:				
15	1	1	1	1
13	0.92 [0.66–1.29]	1.15 [0.87–1.51]	1.46 [1.14–1.87]	1.00 [0.75–1.34]
11	1.43 [0.98–2.08]	2.96 [2.22–3.94]	3.50 [2.69–4.55]	1.11 [0.78–1.58]
Family affluence				
High	1	1	1	1
Medium	0.87 [0.59–1.27]	0.96 [0.71–1.31]	1.02 [0.77–1.34]	0.70 [0.51–0.98]
Low	1.31 [0.83–2.06]	1.54 [1.07–2.20]	1.59 [1.15–2.21]	0.89 [0.59–1.34]
Health literacy				
High	1	1	1	1
Medium	1.10 [0.74–1.62]	1.27 [0.92–1.74]	1.67 [1.25–2.24]	0.54 [0.39–0.75]
Low	1.84 [0.96–3.54]	2.80 [1.63–4.82]	1.83 [1.05–3.19]	0.86 [0.45–1.65]
Academic achievement				
High	1	1	1	1
Medium	1.82 [1.12–2.95]	1.32 [0.94–1.87]	1.05 [0.78–1.42]	1.32 [0.91–1.93]
Low	4.41 [2.62–7.41]	2.06 [1.37–3.11]	1.60 [1.11–2.30]	2.18 [1.39–3.41]
Peer support				
High	1	1	1	1
Medium	1.40 [0.98–1.99]	1.66 [1.25–2.21]	1.43 [1.10–1.85]	0.81 [0.57–1.16]
Low	1.62 [0.89–2.94]	2.74 [1.70–4.44]	1.94 [1.22–3.08]	1.19 [0.66–2.15]
Family support				
High	1	1	1	1
Medium	2.27 [1.58–3.26]	1.13 [0.82–1.55]	1.02 [0.76–1.36]	1.75 [1.25–2.45]
Low	1.61 [0.88–2.96]	0.68 [0.40–1.13]	0.69 [0.43–1.12]	1.11 [0.61–2.04]
Parental monitoring				
High	1	1	1	1
Medium	0.75 [0.50–1.14]	1.04 [0.70–1.46]	0.97 [0.71–1.33]	0.78 [0.52–1.15]
Low	0.61 [0.39–0.95]	1.25 [0.87–1.79]	0.83 [0.60–1.15]	0.77 [0.52–1.14]
Self-rated health				
Good	1	1	1	1
Poor	1.15 [0.78–1.68]	1.39 [1.01–1.90]	1.29 [ 0.97–1.71]	1.26 [0.90–1.76]
Feeling low				
Less than weekly	1	1	1	1
More than weekly	0.81 [0.60–1.09]	0.61 [0.47–0.78]	0.69 [0.55–0.85]	1.20 [0.91–1.58]
Tired				
Less than 4 times a week	1	1	1	1
More than 4 times a week	0.88 [0.65–1.19]	0.91 [0.71–1.17]	0.74 [0.60–0.93]	0.92 [0.70–1.21]
Social media use				
No risk	1	1	1	1
Moderate risk	1.03 [0.76–1.38]	0.43 [0.34–0.55]	0.72 [0.59–0.88]	1.40 [1.06–1.83]
Problematic	3.31 [2.26–4.85]	0.43 [0.27–0.69]	0.45 [0.29–0.69]	2.70 [1.84–3.96]

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
