# Peer review of "Profiles of Internet Use and Health in Adolescence: A Person-Oriented Approach"

_ijerph, 2021, doi:10.3390/ijerph18136972_

Round 1

Reviewer 1 Report

  1. RQ4 presented the question as if health related outcomes are the dependent variable.  However, the analysis was performed with internet user profile as the outcome.  
  2. Please indicate if the result of Table 4 is from a multivariable model.   As it is presented, it looks like a repeat of Table 3.
  3. It would be of interest to the readers to present summary of internet activity by user profile in table format.  The associated figures are hard to follow.  A table with median (IQR) for each activity by user profile may be easier to follow. 
  4. The groups are different in characteristics (table 3/4).  Are there differences between user profiles in any of the internet activities?

Author Response

Dear reviewer,

My responses can be found from the Word file.

Best wishes, Henri Lahti

Reviewer 2 Report

Dear authors

I consider you submitted an article of great interest to the scientific community, having selected an appropriate methodological design, although there are some aspects that must be improved before it get published.

  1. The population has not been specified, it would be neccesary to have this data, in addition to making the calculations around the acceptable sample size considering the context of the research. Also it should be explained why the research only considered persons aged 11, 13, and 15 (12 and 14?).
  2. Regarding the scale used to measure Internet activity, a reference is not included, is it a standardized scale or has it been created ad-hoc? If this is the case, the creation and validation process must be detailed.
  3. Regarding the rest of the existing scales that have been used, the validation process has not been described, and it is very important to know it theses scales are valid in the context to which they have been used (adolescents 11-15). Are there other studies in which these scales have been tested in analogous context? If so, please provide references for that, otherwise a scale validation process for all scales used must be included. On the other hand, for those scales structured in categories, evidence must be provided from previous research studies concluding the factorization in these scales.
  4. In the description of results, table 1 is not mentioned within the text. Beisdes, both in tables 1 and 3, the interpretation given to the calculation of the IC and the chi-square is not provided.
  5. I would suggest dividing figure 1 into two figures, one for the means and the other for the modes, as well as making a distinction of what both calculations contribute to the results.

Regards

Author Response

Dear reviewer,

The responses can be found from the word file attached.

Best wished, Henri Lahti

Reviewer 3 Report

It would be neccessary not to use subjective expressions.

The conclusions could be more explained. They don´t reflect all the research .

Future research must be proposed in the article.

Author Response

Dear reviewer,

The responses can be found in the attached word file.

Best wishes, Henri Lahti

Round 2

Reviewer 2 Report

Dear authors

Thanks for the corrections and respond to my suggestions.

Kind regards